# Metabolic Profiling of Rheumatoid Arthritis Neutrophils Reveals Altered Energy Metabolism That Is Not Affected by JAK Inhibition

**DOI:** 10.3390/metabo12070650

**Published:** 2022-07-15

**Authors:** Susama Chokesuwattanaskul, Michele Fresneda Alarcon, Sangeetha Mangalakumaran, Rudi Grosman, Andrew L. Cross, Elinor A. Chapman, David Mason, Robert J. Moots, Marie M. Phelan, Helen L. Wright

**Affiliations:** 1Institute of Integrative Biology, University of Liverpool, Liverpool L69 7BE, UK; dr_susama_c@yahoo.com; 2Department of Musculoskeletal and Ageing Science, Institute of Life Course and Medical Sciences, University of Liverpool, Liverpool L7 8TX, UK; m.fresneda-alarcon@liverpool.ac.uk (M.F.A.); across@liverpool.ac.uk (A.L.C.); e.chapman@bangor.ac.uk (E.A.C.); 3School of Life Sciences, University of Liverpool, Liverpool L69 7ZB, UK; sangee@hotmail.co.uk; 4Institute of Systems, Molecular and Integrative Biology, University of Liverpool, Liverpool L69 7BE, UK; hlrgrosm@liverpool.ac.uk (R.G.); mphelan@liverpool.ac.uk (M.M.P.); 5High Field NMR Facility, Liverpool Shared Research Facilities University of Liverpool, Liverpool L69 7TX, UK; 6Centre for Cell Imaging, Institute of Integrative Biology, University of Liverpool, Liverpool L69 7TX, UK; dma@visiopharm.com; 7Department of Rheumatology, Aintree University Hospital, Liverpool L9 7AL, UK; robert.moots@liverpoolft.nhs.uk; 8Faculty of Health, Social Care and Medicine, Edge Hill University, Ormskirk L39 4QP, UK

**Keywords:** JAK inhibitors, metabolomics, NMR, rheumatoid arthritis, host defence, neutrophils, NETs

## Abstract

Neutrophils play a key role in the pathophysiology of rheumatoid arthritis (RA) where release of ROS and proteases directly causes damage to joints and tissues. Neutrophil function can be modulated by Janus Kinase (JAK) inhibitor drugs, including tofacitinib and baricitinib, which are clinically effective treatments for RA. However, clinical trials have reported increased infection rates and transient neutropenia during therapy. The subtle differences in the mode of action, efficacy and safety of JAK inhibitors have been the primary research topic of many clinical trials and systematic reviews, to provide a more precise and targeted treatment to patients. The aim of this study was to determine both the differences in the metabolome of neutrophils from healthy controls and people with RA, and the effect of different JAK inhibitors on the metabolome of healthy and RA neutrophils. Isolated neutrophils from healthy controls (HC) (*n* = 6) and people with RA (*n* = 7) were incubated with baricitinib, tofacitinib or a pan-JAK inhibitor (all 200 ng/mL) for 2 h. Metabolites were extracted, and ^1^H nuclear magnetic resonance (NMR) was applied to study the metabolic changes. Multivariate analyses and machine learning models showed a divergent metabolic pattern in RA neutrophils compared to HC at 0 h (F1 score = 86.7%) driven by energy metabolites (ATP, ADP, GTP and glucose). No difference was observed in the neutrophil metabolome when treated with JAK inhibitors. However, JAK inhibitors significantly inhibited ROS production and baricitinib decreased NET production (*p* < 0.05). Bacterial killing was not impaired by JAK inhibitors, indicating that the effect of JAK inhibitors on neutrophils can inhibit joint damage in RA without impairing host defence. This study highlights altered energy metabolism in RA neutrophils which may explain the cause of their dysregulation in inflammatory disease.

## 1. Introduction

Neutrophils are the most abundant leukocyte in humans, produced by the bone marrow at a rate of 5–10 $\times \;10^{10}$ per day. They are specialist cells of the innate immune system that play a major role in host defence against micro-organisms through phagocytosis and generation of reactive oxygen species (ROS). Neutrophils have been shown to have the greatest potential to cause damage to local tissues when dysregulated [1] and are key mediators in the pathology of rheumatoid arthritis (RA), the commonest form of inflammatory arthritis. RA is a chronic autoimmune, inflammatory condition characterized by inflammation of the tendon sheaths (tenosynovitis) and joint lining (synovitis) leading to growth of an inflammatory pannus which quickly erodes the joint cartilage and bone [2]. When improperly activated, neutrophils secrete ROS, degradative enzymes, and inflammatory mediators such as cytokines and chemokines directly onto joint tissue. Neutrophils also expose auto-antigens through the production of neutrophil extracellular traps (NETs) [3,4,5].

New highly effective treatments for RA are orally-available Janus Kinase (JAK) inhibitors, such as tofacitinib (JAK1/JAK3 inhibitor) and baricitinib (JAK1/JAK2 inhibitor) that target the JAK/STAT pathway. JAK inhibitors decrease cytokine-induced JAK activation, and in turn decrease the activation of intracellular STAT transcription factors that mediate many aspects of cellular immunity [6,7,8,9,10]. Clinical trials of both tofacitinib and baricitinib have reported a transient drop in neutrophil counts during therapy [11,12,13] and increased rates of infection with serious implications for the host in immune-suppressed patients [7,8,13,14,15]. It has been suggested that tofacitinib inhibits GM-CSF–induced Janus kinase 2 (Jak2)-mediated signal transduction, and it completely abrogated GM-CSF-induced IL-1β and caspase-1 (p20) secretion from neutrophils by inhibiting NLRP3 protein expression [16]. However, a greater understanding of the roles of JAK/STAT signalling, and its importance in neutrophil activation is required to fully understand the heterogeneity and functional significance of JAK/STAT inhibition in RA.

The role of cellular metabolism in the context of immunity and inflammation has increased the understanding of immunological processes, and fine-tuning of metabolism during an inflammatory response is key for resolution [5]. Dysregulation of metabolic control has been identified in inflammatory diseases such as RA [17]. Metabolic profiling of immune cells has been achievable thanks to the advances in ${}^{1}$H-NMR spectroscopy, which allows for simultaneous detection and annotation of multiple metabolites, providing quantitative biochemical information [18,19]. Application of NMR metabolomics to neutrophils could help characterise physiological changes associated with inflammatory disease and identify novel therapeutic targets.

The aim of this study was to compare the metabolome of neutrophils from healthy controls (HC) with that of people with RA to determine underlying neutrophil metabolic differences that would provide new insights into the physiology of inflammatory neutrophils. We also sought to determine the effect of therapeutic JAK inhibitors on key neutrophil metabolites, metabolic pathways and inflammatory functions. Using ^1^H-NMR coupled with multivariate statistical analysis, we show that we can classify RA neutrophils from healthy controls based on their metabolic profile. Furthermore, we determine metabolic and functional changes in neutrophils when treated with different JAK compared to untreated controls.

## 2. Materials and Methods

### 2.1. Patient Demographics

This study was approved by the University of Liverpool Central University Research Ethics Committee C for HC, and NRES Committee North West (Greater Manchester West, Manchester, UK) for RA patients. All participants gave written, informed consent in accordance with the declaration of Helsinki. All patients fulfilled the ACR 2010 criteria for the diagnosis of RA [20] and were Biologics naïve. People with RA (age 22–84, 74% female) were recruited from University Hospital Aintree and Broadgreen Hospital in Liverpool. Healthy controls (age 27–58, 53% female) were recruited from colleagues at the University of Liverpool.

### 2.2. Neutrophil Preparation

Whole blood was collected into lithium-heparin vacutainers and within 15 min, mixed with HetaSep solution at a ratio of 1:5 (HetaSep:whole blood) and incubated at 37 ${}^{\circ }$C for 30 min until the plasma/erythrocyte interphase was at approximately 50% of the total volume. Nucleated cells were collected and layered on top of Ficoll–Paque solution at a ratio of 1:1, and then centrifuged at 500× *g* for 30 min. The peripheral blood mononuclear cell (PBMC) layer, plasma, and Ficoll–Paque solution were carefully removed, leaving a neutrophil pellet (purity typically >97%) [21,22]. Pellets were re-suspended in RPMI 1640 media including L-glutamine at a concentration of 5 × 10${}^{6}$ cells/mL. Neutrophils were either left unstimulated or treated with therapeutically relevant concentrations of tofacitinib (200 ng/mL), baricitinib (200 ng/mL) [11,23] or pan-JAK inhibitor I (200 ng/mL) and incubated for 2 h. Deuterated DMSO was used as a vehicle control in all incubations at the same concentration as JAKi (*v*/*v*).

### 2.3. Intracellular Metabolite Extraction and NMR Processing

Neutrophils were prepared for metabolite extraction following our optimised protocol for human neutrophils [24]. Briefly, cells were centrifuged at 1000× *g* at 25 ${}^{\circ }$C for 2 min. The supernatant was aspirated, and cell pellets were re-suspended with ice cold PBS, then centrifuged at 1000× *g* at 25 ${}^{\circ }$C for 2 min. The supernatant was discarded, while the pellets were heated at 100 ${}^{\circ }$C for 1 min, and then snap-frozen in liquid nitrogen. All samples were stored at −80 ${}^{\circ }$C prior to intracellular metabolite extraction. Metabolites were extracted by addition of 50:50 *v*/*v* ice cold HPLC grade acetonitrile:water at 500 µL per cell pellet, followed by a 10 min incubation on ice. Then, samples were sonicated three times for 30 s at 23 kHz and 10 µm amplitude using an exponential probe, with 30 s rest between sonications in an ice water bath. Sonicated samples were centrifuged at 12,000× *g* for 5 min at 4 ${}^{\circ }$C and the supernatant transferred to cryovials, flash frozen in liquid N_2_ and lyophilised [25]. Each lyophilised sample was resuspended in 200 µL of 100 µM deuterated sodium phosphate buffer pH 7.4, with 100 µM trimethylsilyl propionate (TSP) and 0.05% NaN_3_. Each sample was vortexed for 20 s and centrifuged at 12,000× *g* for 1 min at 20 ${}^{\circ }$C. Then 180 µL of each cell extract sample was transferred to 3 mm (outer diameter) NMR tubes for acquisition.

### 2.4. ${}^{1}$H-NMR Measurements

The samples were analysed using a 700 MHz NMR Avance IIIHD Bruker NMR spectrometer equipped with a TCI cryoprobe. Samples were referenced to trimethylsilylpropanoic acid (TSP) at 0 ppm. Spectra was acquired at 25 ${}^{\circ }$C using the 1D Carr–Purcell–Meiboom–Gill (CPMG) edited pulse sequence technique with 512 scans. The spectra were assessed to conform to minimum quality criteria as outlined by the Metabolomics Society [26] to ensure consistent linewidths, baseline corrections and water suppression. All spectra passing quality criteria were then divided into “bins” that were defined globally by the peak limits using Chenomx NMR Suite 7.1 (Chenomx Inc., Edmonton, Alberta, Canada) [27]. All peaks, both annotated in Chenomx (via manual analyses in TopSpin and Chenomx software) and unknown, were included in the bin table. A correlation-based scoring (CRS) method developed by Grosman [28] was applied to the data which aimed at addressing the problem of selecting appropriate representative bins from feature extraction in multivariate analysis. A list of representative bins per metabolite identified was obtained (Appendix A), and statistical analysis was carried out. Metabolomics data have been deposited into the EMBL-EBI MetaboLights database [29] with the identifier MTBLS4766. The complete dataset can be accessed here: www.ebi.ac.uk/metabolights/MTBLS4766.

### 2.5. Intracellular ROS Production in Response to fMLP

Neutrophils (5 × 10${}^{6}$/mL) and were incubated with or without tofacitinib, baricitinib or pan-JAK inhibitor I (all 200 ng/mL) for 30 min prior to GM-CSF (5 ng/mL) priming for 45 min. Dihydrorhodamine-123 (DHR123, 5 µM) was added for 15 min along with fMLP (1 µM) to stimulate ROS production. DHR123 fluorescence in response to ROS was measured using a Beckman Coulter CytoFLEX flow cytometer, with 10,000 events analysed.

### 2.6. Bacterial Killing Assay

*S. aureus* (Oxford strain) were grown from a single colony and adjusted to a final concentration 10${}^{8}$ cells/mL in PBS. Bacteria were opsonised with 30% human AB serum in PBS for 30 min at 37 ${}^{\circ }$C in a shaking incubator. Opsonised bacteria were washed three times in 2 mL PBS and resuspended in PBS (10${}^{9}$ cells/mL). Neutrophils (5 × 10${}^{6}$/mL) were incubated with or without tofacitinib, baricitinib or pan-JAK inhibitor I (all 200 ng/mL) for 30 min, and 10${}^{6}$ cells removed. Neutrophils were incubated for 1 h with 10${}^{7}$*S. aureus* in a shaking incubator. In addition, 10${}^{7}$*S. aureus* were also added to 200 µL RPMI as a positive control. Neutrophils were lysed in 20 mL of deionised water by vortex for 20 s. This was further diluted 1:10 with deionised water, and 50 µL was spread on triplicate LB agar plates and incubated at 37 ${}^{\circ }$C. Colonies were counted after 24 h incubation.

### 2.7. ROS Production in Response to Live S. aureus

Neutrophils (2.5 × 10${}^{5}$) were incubated with 10${}^{7}$ serum-opsonised *S. aureus* (Oxford strain) in the presence of 10 µM luminol, and luminol-enhanced chemiluminescence was measured continuously for 60 min on a Tecan plate reader.

### 2.8. Visualisation of NET Production by Immuno-Histochemistry

Neutrophils were seeded (2 × 10${}^{5}$ cells/500 µL) in RPMI media plus HEPES plus 2% AB serum in a 24-well plate containing poly-L-lysine-coated glass coverslips. Cells were allowed to adhere for 1 h prior to incubation with or without tofacitinib, baricitinib or pan-JAK inhibitor I (all 200 ng/mL) for 30 min. Cells were left unstimulated or stimulated with phorbol 12-myristate 13-acetate (PMA, 100 nM) and incubated for a further 4 h to allow for NET production. Cells adhered to coverslips were fixed with 4% paraformaldehyde prior to immuno-histochemical staining. Briefly, coverslips were removed from the plate and washed with PBS, permeabilised with 0.05% Tween 20 in TBS, blocked with TBS (2% BSA). Primary antibodies used were rabbit anti-neutrophil elastase (1:200) and mouse anti-myeloperoxidase (1:1000). Coverslips were washed three times with TBS prior to secondary antibody staining (anti-rabbit AlexaFluor488, 1:2000 or anti-mouse AlexaFluor647, 1:2000) in TBS (+2% BSA) for 30 min. Coverslips were washed prior to staining with DAPI (1 µg/mL) for 3 min. Coverslips were washed a further three times and mounted onto glass slides using Mowiol 4-88. Images (at least 9 fields per slide) were taken on an Epifluorescent microscope (Zeiss) using the 10X objective by a technician blinded to the experimental conditions. The DAPI channel of one image from each condition was used to blindly train a machine learning pixel classifier in Ilastik v1.3.0 [30] to recognise three categories: background, compact nuclei and NETs. Subsequently, all images in the dataset were processed to produce a “Simple Segmentation” count mask output. A Fiji [31] script was used to measure the area occupied by each label (available at https://bitbucket.org/snippets/davemason/5edXBB).

### 2.9. Quantitative Measurement of DNA Released during NETosis

Neutrophils were incubated in parallel experiments to those described above. At the end of the incubation, 5 µL 0.1 M CaCl_2_ was added to culture supernatant followed by 50 mU micrococcal nuclease and incubated for 10 min at 37 ${}^{\circ }$C. The nuclease reaction was stopped by the addition of 5 µL EDTA (0.5M). Culture supernatants were removed from each well, centrifuged at 200× *g* for 5 min to remove cellular debris, and decanted into clean tubes prior to freezing at −80 ${}^{\circ }$C. DNA content of each supernatant was measured using the Quantifluor dsDNA kit in black 96-well plates using serially diluted lambda DNA as a calibration standard (0–2000 ng/mL). Measurement was carried out at 485 nm excitation/535 nm emission on a Tecan plate reader.

### 2.10. Statistical Analysis

Statistical analyses were performed using R v4.0.2 [32] and the mixOmics package [33]. Metabolomics data were normalised by probabilistic quotient normalization [34,35] and tested for normality with the Shapiro–Wilk test due to the small sample size. Univariate analysis was carried out by ANOVA when comparing more than 2 groups or Student’s *t*-test with application of a False-Discovery Rate (FDR) and adjusted *p*-value of 0.05. For multivariate analysis, the data were normalized and then Pareto scaled before applying unsupervised principal component analysis (PCA). Partial Least Squares Discriminant Analysis (PLS-DA) was used to build predictive models between experimental groups, and model performance was assessed by dividing the data into 70% and 30% training and validation sets, respectively. The training set was cross-validated with the leave-one-out method, and classification errors were used to determine the optimal model complexity parameter. The refined model was then used to predict the validation set to obtain model performance and assessed by calculating accuracy, precision, recall and F1-score [36,37]. Variable importance in projection for each model was extracted to determine each metabolite contribution to the model.

## 3. Results

A total of 54 unique metabolites were annotated from 505 spectral bins across the ^1^H-NMR spectra of neutrophils. Some metabolites such as glucose were represented by multiple spectral bins and identities confirmed where possible. Neutrophil metabolites annotated included amino acids, ketone bodies and several glycolytic intermediates as well as other metabolites. A representative bin for each metabolite was selected using in-house criteria determined by correlation reliability score [28] and used to perform statistical analysis (Appendix A). All metabolite annotation and identities are available via public repository MetaboLights (ID number: MTBLS4766).

### 3.1. Changes in Neutrophil Metabolome Associated with RA

Principal Component Analysis (PCA) was performed to reduce the dimensionality of the metabolomics datasets to evaluate differences in the metabolome between HC subjects and people with RA. Untreated 0 h neutrophils separated into two distinct clusters indicating that the metabolic profile of RA neutrophils is clearly distinct from that of healthy individuals (Figure 1A). Independent samples *t*-test comparing the groups at 0 h found 12 metabolites to be significantly different between RA and HC neutrophils (adj *p*-value < 0.05, Figure 1B). Taurine, ATP, ADP, GTP and glutathione were all significantly increased in RA neutrophils (Figure 1G). Glucose was 2-fold higher in HC neutrophils compared to RA, although this difference was not statistically significant (adj *p*-value = 0.167) highlighting a high variation in glucose content in HC individuals (Figure 1G). The PCA separation between the two groups was maintained after 2 h incubation with and without different JAKi treatments (Figure 1D, and Appendix A), with taurine and energy producing metabolites ATP, ADP always significantly different between the two groups. Partial Least Squares Discriminant Analysis (PLS-DA) was performed to determine which metabolites were responsible for the discrimination between RA and HC, and also to investigate the diagnostic potential with a predictive model. Based on best practices [38], the model validation was repeated multiple times with random test and train data splits to account for the small number of samples and selection bias, to give an average predictive metric of accuracy, balanced accuracy, precision, recall and F1 score for each model (Appendix A). PLS-DA comparing RA and HC neutrophils at baseline 0 h discriminated the two groups (balanced accuracy 90% (13.7σ), precision 100% ± (0σ), F1 score 86.7% (18.3σ)). The most influential metabolites in the classification (VIP > 1) were taurine, D-glucose, phosphocholine and formic acid (Figure 1C). The same metabolites are the most important in the model constructed from the 2 h untreated comparison (Figure 1F).

### 3.2. Changes in Neutrophil Metabolome after 2 h Incubation

HC and RA neutrophils were incubated without JAK inhibitor treatment for 2 h before extraction of intracellular metabolites. HC and RA neutrophils were analysed separately by PCA, and each group showed a clear separation by timepoint (0 h to 2 h) on PC1 (Figure 2A,C). The significant changes in neutrophil metabolome after incubation for 2 h were determined by paired sample *t*-test. HC neutrophils had 31 metabolites significantly different after 2 h (adj *p*-value < 0.05, Figure 2B,E) whereas only 12 were significant after 2 h in RA neutrophils (adj *p*-value<0.05, Figure 2D,E). Energy production metabolites ATP and ADP were significantly different in both RA and HC neutrophils, decreasing after 2 h incubation (Figure 2F). Taurine and glutathione significantly decreased in RA neutrophils only after 2 h (Figure 2F) with respect to 0 h. Intracellular glucose levels increased in both RA and HC neutrophils, but only reached statistical significance in HC (adj *p*-value < 0.05). Both leucine and valine (branched chain amino acid) increased significantly after 2 h incubation (Figure 2F).

### 3.3. Changes in Neutrophil Metabolome Following Treatment with JAK Inhibitors

To test the differences in the neutrophil metabolome induced by JAK inhibitors, we incubated RA and HC neutrophils with or without baricitinib, tofacitinib or a pan-JAK inhibitor for 2 h. Following metabolite extraction, one-way ANOVA on all 2 h JAK inhibitor samples was used to compare the 2 h treated and untreated samples. In both HC and RA neutrophils, no significant differences were observed. Tukey’s post-hoc analysis revealed no metabolites with an adj *p*-value < 0.05 for any pairwise comparison. PCA revealed a high between-subject variability (Appendix A), which is the dominant feature and potentially masks the underlying effects of JAK inhibitors on the neutrophil metabolome. Subsequent analysis by paired *t*-test directly compared 0 h untreated neutrophils with the 2 h JAK inhibitor treated samples (Figure 3). Direct comparison of individual metabolites shows a very similar profile in HC neutrophils when left untreated or treated with baricitinib or tofacitinib. HC neutrophils treated with pan-JAK inhibitor showed only a significant difference for ATP. RA neutrophils treated with baricitinib or a pan-JAK inhibitor had a similar profile to the 2 h untreated condition when compared to untreated 0 h, with metabolites associated in energy metabolism (ATP and ADP) significant across these conditions.

### 3.4. Effect of JAK Inhibitors on ROS and NET Production

Our univariate analyis of HC and RA neutrophil metabolomes identified energetic metabolites such ATP and ADP as key metabolites that exhibit dynamic changes in abundance in neutrophils. Furthermore, we identified metabolites such as NAD and NADP+ to be consistently between 1.5 and 3 folds higher in RA compared to HC neutrophils at 0 h. After incubation with JAK inhibitors, these metabolites are consistently between 4 to 8 folds higher in RA compared to HC (Figure 4).

NAD and NADP+ are key components of the pentose phosphate pathway, important in neutrophils for the production of reactive oxygen species (ROS) and neutrophil extracellular traps (NETs), both of which are implicated in damage to joints and tissues in RA [39]. We therefore investigated the effect of JAKi treatment on ROS and NET production by HC and RA neutrophils. This investigation was carried out with a new cohort of people with RA (*n* = 20; *n* = 10 DMARD naïve, *n* = 10 Biologics naïve) and HC (*n* = 10). We did not detect any significant functional differences in RA neutrophils based on whether they were DMARD-naive or Biologic-naive (data not shown). ROS production was measured in GM-CSF-primed neutrophils, in response to the bacterial peptide fMLP. Neutrophils were treated with JAK inhibitors baricitinib, tofacitinib and a pan-JAK inhibitor for 30 min prior to priming for 45 min with GM-CSF. ROS production was stimulated by fMLP (10${}^{-3}$), and measured by DHR123, which emits fluorescence when excited by intracellular H_2_O_2_-derived ROS (and mitochondrial oxidants) [40]. All three JAK inhibitors significantly decreased the percentage of DHR123 positive HC and RA neutrophils (Figure 5A, *p*-value < 0.05). We also measured ROS production in response to live, opsonised *Staphylococcus aureus* bacteria using luminol-enhanced chemiluminescence. Luminol emits light upon excitation by ROS, measuring both intra- and extra-cellular ROS production by myeloperoxidase in the neutrophil respiratory burst [40]. Unprimed neutrophils were incubated with *S. aureus* for 60 min in a plate reader, and luminescence was read continuously. None of the inhibitors tested significantly decreased the amount of ROS measured in response to phagocytosis of *S. aureus* (Figure 5B). Finally, we investigated the effect of JAK inhibitors on the ability of RA and HC neutrophils to kill live, opsonised *S. aureus* over 90 min. Whilst the mean number of bacteria killed by both RA and healthy neutrophils was decreased by all three JAK inhibitors, these numbers did not reach statistical significance (*p*-value > 0.05) (Figure 5C). None of the JAK inhibitors had any significant effect on chemotaxis or phagocytosis of FITC-labelled latex beads (Appendix A).

As an alternative to phagocytosis and cytotoxic killing of bacteria, neutrophils may release NETs to trap and kill pathogens. However, in auto-immune diseases, the externalisation of NET DNA and proteins may contribute towards the formation of auto-antibodies by exposing intracellular epitopes to the immune system [41]. In order to determine the effect of the three JAK inhibitors on NET production (NETosis), we pre-incubated neutrophils with JAK inhibitors for 30 min and then incubated neutrophils for a further 4 h either unstimulated, to measure spontaneous levels of NETosis, or with PMA, a potent activator of protein kinase C, or TNFα.

Both PMA and TNFα have been reported in the literature to stimulate NETosis in RA neutrophils [41]. However, we found that TNFα did not significantly induce NET formation in healthy or RA neutrophils (data not shown), and there was no increase in the level of spontaneous NETosis in RA compared to healthy controls, in line with previous observations from experiments in our hands [42] (Figure 6A,B). PMA significantly increased the levels of externalised NET DNA in culture supernatants compared to untreated neutrophils both in RA (Figure 6A *p*-value < 0.001) and HC (*p*-value < 0.05). None of the JAK inhibitor treatments significantly decreased the amount of DNA released by PMA-stimulated neutrophils. In addition to quantification of externalised DNA in culture supernatants, we used machine learning to segment DAPI-stained images into three classes: background, compact nuclei and NETs as previously described [43]. PMA significantly increased NET production by both HC and RA neutrophils (Figure 6B,C *p*-value < 0.001). Baricitinib treatment decreased the level of NETs produced by both PMA-stimulated RA (*p*-value < 0.01) and HC neutrophils (*p*-value < 0.05). Tofacitinib increased NET production by both RA and HC neutrophils (*p*-value < 0.05 in HC) and pan-JAK inhibitor I treatment significantly decreased NET production by HC but not RA neutrophils (JAKi *p*-value < 0.001).

## 4. Discussion

Metabolomics is emerging as a tool to identify biomarkers for disease, response to treatment and also indicators of pathogenesis that may inform routes for novel interventions [44]. In this study, we applied ${}^{1}$H-NMR metabolomics to determine the variances in the metabolome of neutrophils from HC and people with RA, with or without treatment with drugs targeting JAKs. Previous studies have shown detectable differences in the metabolome of biofluids from RA and HC, including urine and plasma [45,46]. Here, we have described the first comparative ${}^{1}$H-NMR-based metabolomics investigation comparing HC and RA neutrophils from whole blood. We detected a total of 53 metabolites in RA and HC neutrophils which were a combination of amino acids, fatty acids, sugars, purines and carboxylic acids. Using PLS-DA models, we were able to classify RA and HC neutrophils with a high degree of accuracy based on metabolite abundances from NMR analysis.

Neutrophils are known to meet their energy needs by utilising the glycolytic pathway [47]. Their reliance on glycolysis is necessary to enable responses including migration, pathogen clearance, and apoptosis. Our data confirm an energetic imbalance in RA neutrophils with a more metabolically active phenotype in RA demonstrated by the increase in abundance of energy related metabolites such as ADP and ATP. The NMR analysis shows that metabolites closely related to the activation of the NADPH oxidase (NOX2) complex, such as NADP+ and NAD are consistently increased in RA at 0 h and in all treatment conditions after 2 h incubation. Neutrophils produce ROS via activation of NOX2 and in RA; both blood and synovial fluid neutrophils have an increased capacity to produce ROS [48]. Furthermore, glutathione and taurine were significantly increased in RA neutrophils. Taurine is the most abundant free amino acid in humans, and it is known to be the primary molecule to react with and detoxify hypochlorous acid (HOCl) produced by the neutrophil myeloperoxidase (MPO), forming a less toxic taurine chloramine [49,50]. It has been shown that taurine enhances expression and activation of antioxidant enzymes, such as superoxide dismutase, catalase and glutathione peroxidase [49,51]. Taurine is also significantly decreased in aged, mainly apoptotic neutrophils [52]. Chemically reduced glutathione is crucial for the detoxification of hydrogen peroxide (H_2_O_2_) produced by NOX2 assembled on the membrane [53]. These overlapping results highlight the importance of detoxifying agents in neutrophil viability [54]. These increased or decreased metabolites are not necessarily pathogenic, but a biomarker of an altered metabolic pathway. The increased metabolic activity paired with the significantly elevated detoxification metabolites in RA neutrophils suggests a metabolic adaptation of RA neutrophils to cytosolic acidification caused by the constant activation state in RA neutrophils.

Incubation of neutrophils in culture media for 2 h showed a general increase of glucose in both RA and HC neutrophils which may be correlated to the relatively high availability of this metabolite in culture media (around 10 mM). However, ATP and ADP decreased both in RA and HC neutrophils meaning that the energy producing pathways were not sustained during incubation despite the high abundance of glucose. In both RA and HC groups, the most significantly increased metabolites after 2 h in culture were amino acids. However, the increase in uptake of leucine, glutamine and valine was much greater in HC neutrophils. These amino acids are all media components which may be expected to appear in the NMR spectra of neutrophils following incubation in culture media. However, the difference in uptake between RA and HC neutrophils was not expected and may reflect important differences in uptake and breakdown of these metabolites by neutrophils.

The final aim was to determine metabolic differences in neutrophils treated with JAK inhibitors. JAK tyrosine kinases are bound to the cytoplasmic regions of membrane receptors, which respond to agonists including cytokines and growth factors [15,55]. Specific combinations of different JAKs induce a wide-range of signalling cascades (JAK/STAT signalling), and the JAK/STAT pathway is unmatched among known signalling cascades for variety and gene expression [56]. The array of STAT dimerization increases the range of gene-specific binding sites, contributing to the efficiency of nuclear translocation and varied biologic responses [57]. Targeting specific JAK heterodimers could potentially distinguish the individual efficacy and safety profiles of therapeutic JAK inhibitors [58]. Tofacinib and baricitinib are two therapeutic JAK inhibitors that target different receptor heterodimers, and our original hypothesis was that inhibiting a specific JAKs may impact multiple metabolic pathways, explaining both the efficacy and adverse effects observed with JAK inhibitors [59]. However, in this study, we were not able to identify significant metabolic differences between different JAK inhibitor treatments in this modest sample size. Clinical trials of JAK inhibitors have identified a significant neutropenia that is associated with a significantly increased risk of infection. As part of our study, we tested the effect of JAK inhibitors on neutrophil ROS production and NETosis. NETosis was significantly decreased in both groups by baricitinib but not tofacitinib, and intracellular ROS production measured by DHR123 was significantly decreased in both RA and HC neutrophils compared to the GM-CSF-primed neutrophils. We have previously shown that JAK inhibition by either baricitinib or tofacitinib significantly reduces cytokine-induced STAT activation. Baricitinib and tofacitinib also abrogate interferon-γ or GM-CSF delayed apoptosis in HC neutrophils and decrease the levels of STAT phosphorylation in RA neutrophils [22]. JAK inhibition also significantly decreases random RA neutrophil migration and GM-CSF priming of ROS production in HC neutrophils [22]. Importantly in this study, killing of *S. aureus* bacteria was not impaired by JAK inhibitors, suggesting the effects of baricitinib and tofacitinib on neutrophil activation and ROS production may be beneficial in cytokine-driven inflammatory diseases such as RA but not detrimental to neutrophil host defence and bacterial killing within the phagosome.

One of the limitations of our study is that the effect of JAK inhibitor treatment on neutrophil metabolism was measured in vitro after 2 h incubation with inhibitors, and not in vivo after oral administration of therapeutic JAK inhibitors. Therefore, the changes reported in our study may not fully represent the changes that take place in vivo during JAK inhibitor therapy. We have previously shown that baricitinib and tofacitinib are rapidly taken up by RA neutrophils in cell culture where they not only prevent cytokine-induced phosphorylation of transcription factors, but also reverse cytokine-priming of ROS production in a little as 30 min [22]. Therefore, their full effect on neutrophil metabolism should be evident after 2 h incubation. Future studies should extend this work via a longitudinal study of people with RA pre- and post- oral administration of baricitinib and tofacitinib to confirm the in vivo effects on metabolism, ROS and NET production described in this work, and establish how this correlates to improvements in disease activity. Such clinical studies will also account for the effect of JAK inhibition on other elements of the immune system, which will likely impact neutrophil phenotype. The JAK-STAT pathway plays a role in development, proliferation and function of T, B and NK cells. These cells produce and respond to cytokines including IL-2, IL-4, IL-7, IL-9, IL-15 and IL-21 which rely on JAK1-JAK3 activity [15,60,61,62]. In the context of RA, the effects of JAK-STAT signalling include but are not limited to production of the chemokine IL-10 by collagen-stimulated B cells [63] which inhibits neutrophil recruitment [64] and IL-9 production by Th-9 cells, prolonging the survival of neutrophils in synovial fluid and increasing MMP-9 production [65,66].

In summary, this study has described for the first time key differences in the metabolite profiles of HC and RA neutrophils, including differences in metabolites involved in energy and ROS production. We have also described key differences in metabolite profiles of RA and HC neutrophils following culture in vitro for 2 h, which may be attributed to increased metabolic activity in RA neutrophils and differences in the import and/or turnover of metabolites from culture media. Finally, whilst JAK inhibitors did not significantly alter the metabolome of RA or HC neutrophils, we showed that therapeutic JAK inhibitors baricitinib and tofacitinib significantly inhibited ROS and NET production associated with inflammatory activation but did not inhibit bacterial killing important for host defence. We believe that dysregulated neutrophil metabolism is a novel signalling mechanism that could be therapeutically targeted to reset the immune system in inflammatory disease, and that ^1^H-NMR metabolomics is a promising technique for molecular fingerprinting in a clinical setting to aid diagnostics and treatment stratification.

## Figures and Tables

**Figure 1 metabolites-12-00650-f001:**
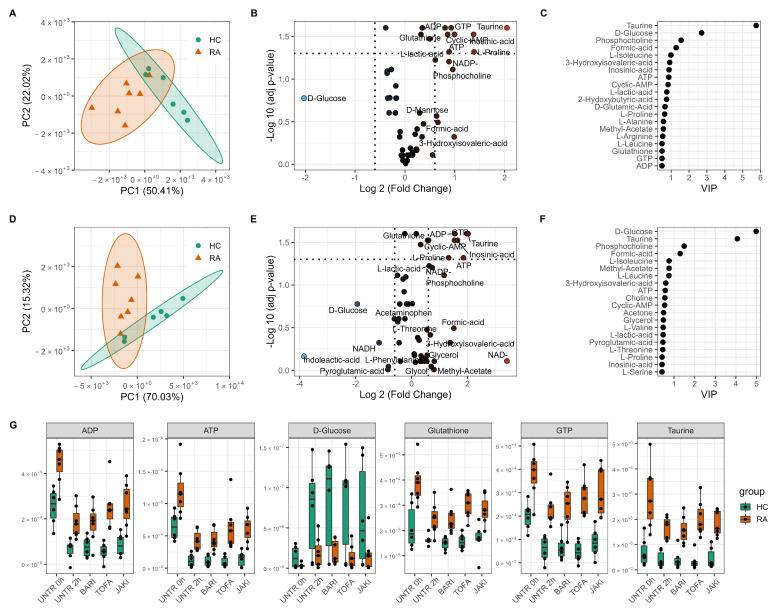
Metabolomics differences between rheumatoid arthritis (RA) and healthy control (HC) neutrophils. PCA scores plot of HC and RA neutrophils at 0 h (**A**) and 2 h (**D**) showing separation on the first principal component (PC). Volcano plot (**B**,**E**) showing metabolites significantly different between HC and RA neutrophils (adj *p*-value < 0.05) and the log2 fold change (FC) for each metabolite as indicated by gradient colour scale provided. Variable importance in projection (VIP) (**C**,**F**) obtained from PLS-DA showing top 20 metabolites for each model. (**G**) boxplot of selected metabolites varying between HC and RA neutrophils with JAK inhibitor treatments (UNTR = Untreated, BARI = Baricitinib, TOFA = Tofacitinib, JAKi = Pan-Jak inhibitor).

**Figure 2 metabolites-12-00650-f002:**
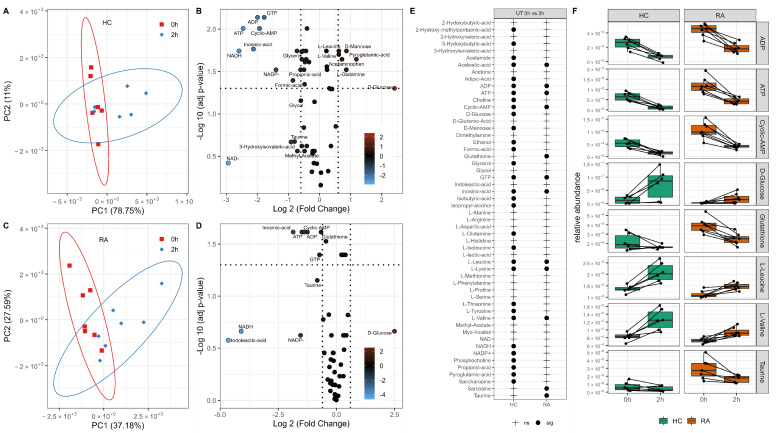
Metabolic adaptation of neutrophils after in vitro incubation. PCA of neutrophils showing the separation by incubation time in HC (**A**) and RA (**C**). Volcano plots (**B**,**D**) showing metabolites significantly different between 0 h and 2 h incubation (adj *p*-value < 0.05) and the log2 fold change (FC) for each metabolite as indicated by gradient colour scale provided; (**E**) table comparing metabolites are significantly different after in vitro incubation for HC and RA neutrophils. (**F**) Boxplots of significant metabolites as determined from the univariate analysis (adj *p*-value < 0.05).

**Figure 3 metabolites-12-00650-f003:**
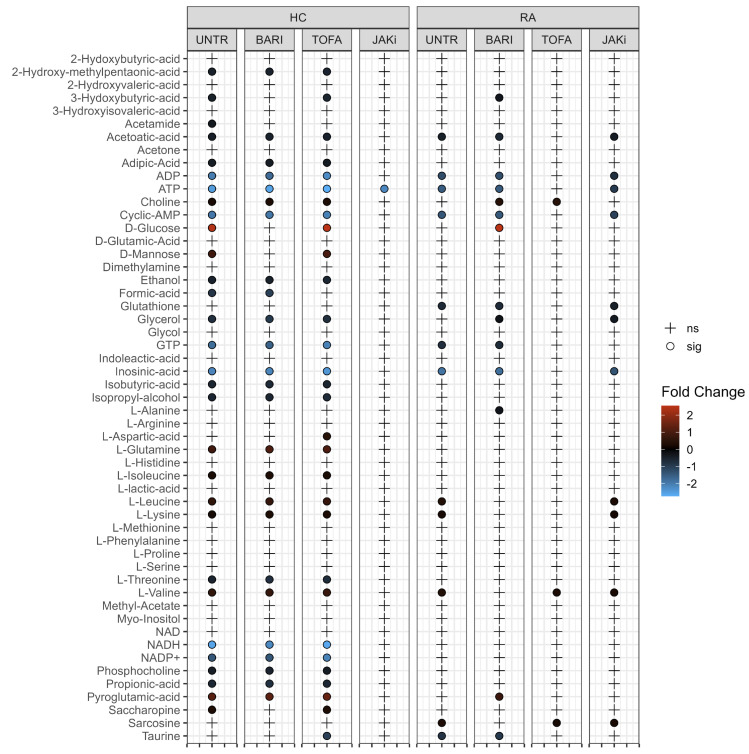
Univariate analysis comparing 0 h untreated HC and RA neutrophil metabolites with each 2 h treatment. (UNTR = Untreated, BARI = Baricitinib, TOFA = Tofacitinib, JAKi = Pan-Jak inhibitor). Metabolites which are significantly different between treatments are indicated by “•” and the log2 fold change (FC) against 0 h control for each metabolite is indicated by gradient colour scale provided.

**Figure 4 metabolites-12-00650-f004:**
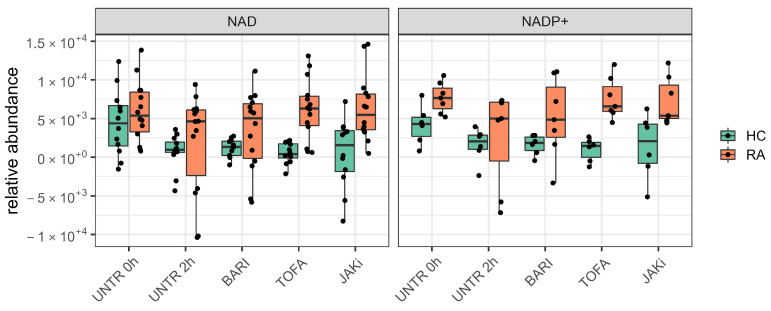
Boxplots comparing abundance of NAD and NADP+ in HC and RA neutrophils across all conditions tested. (UNTR = Untreated, BARI = Baricitinib, TOFA = Tofacitinib, JAKi = Pan-Jak inhibitor).

**Figure 5 metabolites-12-00650-f005:**
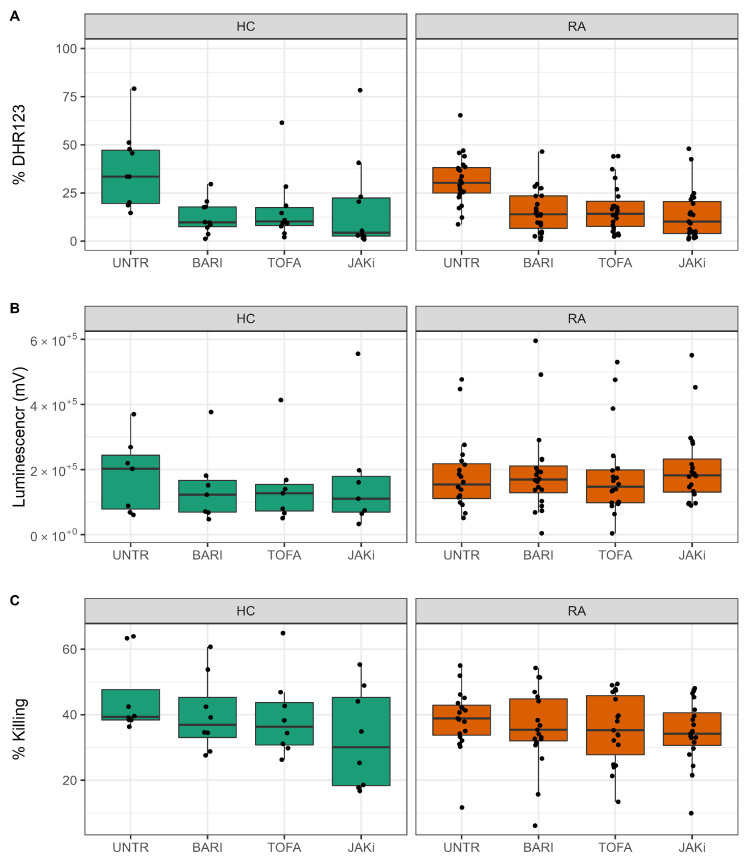
Effect of JAK inhibitors on neutrophil ROS production and bacterial killing. (**A**) All JAK inhibitors decreased ROS production by GM-CSF-primed neutrophils (* *p*-value < 0.05). (**B**) JAK inhibitors did not significantly decrease the amount of ROS produced in response to live opsonised *S. aureus*. (**C**) JAK inhibitors did not significantly decrease bacteria killing by healthy or RA neutrophils. UNTR = untreated, BARI = baricitinib, TOFA = tofacitinib, JAKi = pan-JAK inhibitor. HC (*n* = 10, green), RA (*n* = 20, orange).

**Figure 6 metabolites-12-00650-f006:**
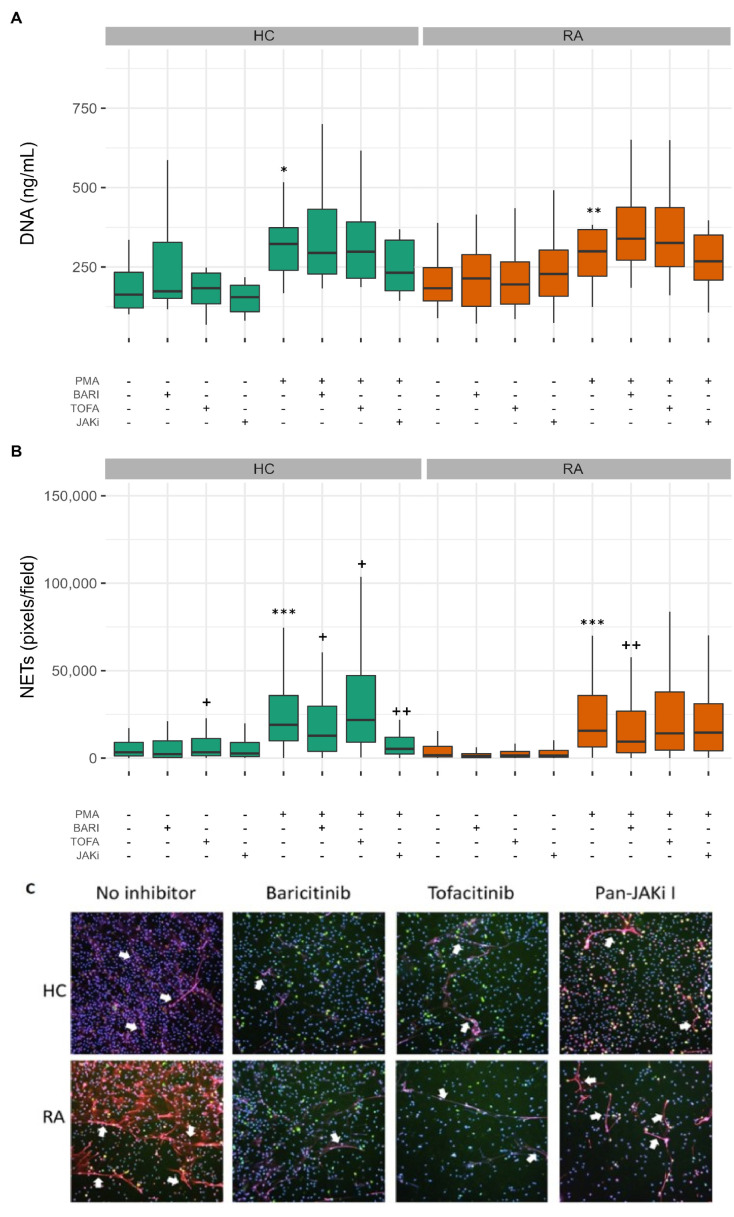
Effect of JAK inhibitors on NET production by HC and RA neutrophils. (**A**) PMA significantly increased the release of NET DNA into culture supernatants by HC and RA neutrophils (* *p*-value < 0.05, ** *p* < 0.01). This was not significantly affected by JAK inhibitors. (**B**) PMA significantly increased NET staining on coverslips (*** *p* < 0.001 compared to untreated). Baricitinib and pan-JAK inhibitor significantly decreased NET production by PMA treated neutrophils whereas tofacitinib increased NET production (+ *p* < 0.05, ++ *p* < 0.01). Machine learning was used to classify pixels as background, compact or NET. (**C**) Representative images for PMA-treated neutrophils are shown. Cells on cover slips were stained for DNA (DAPI, blue), myeloperoxidase (red) and elastase (green). White arrows indicate NET structures. (UNTR = Untreated, BARI = Baricitinib, TOFA = Tofacitinib, JAKi = Pan-Jak inhibitor).

## Data Availability

Metabolomics data have been deposited to the EMBL-EBI MetaboLights database [29] with the identifier MTBLS4766. The complete dataset can be accessed here: https://www.ebi.ac.uk/metabolights/MTBLS4766.

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
