# Peer review of "Metabolic Profiling of Rheumatoid Arthritis Neutrophils Reveals Altered Energy Metabolism That Is Not Affected by JAK Inhibition"

_metabolites, 2022, doi:10.3390/metabo12070650_

Round 1
Reviewer 1 Report
This report provide a clear report on key differences in the metabolite profiles of healthy (HC) and rheumatoid arthritis (RA) neutrophils by 1H nuclear magnetic resonance, and also showed JAK inhibitors did not significantly alter the metabolome and host defenceof RA or HC neutrophils.
The article is well written and data presentation is sound. However, I think the inhibitor treatment on the purified neutrophil for 2 hour in vitro is hard to represent their true effects in vitro, and cannot mimic the neutrophil reactions after real JAK inhibitor treatments of RA patients. In addition, the model author used in this study completely ignored the interactions between other immune cells and the microenvironment of RA. I wish authors can discuss these points in the discussion.
Author Response
Reviewer 1 asks for additional points to add to the discussion (1) suitability of model compared to in vivo effect of JAKi therapy (2) interactions with other immune cells in microenvironment.
We thank the reviewer for this suggestion and we have now added the following paragraph to the manuscript discussion (lines 389-408)
“One of the limitations of our study is that the effect of JAK inhibitor treatment on neutrophil metabolism was measured in vitro after 2h incubation with inhibitors, and not in vivo after oral administration of therapeutic JAK inhibitors. Therefore, the changes reported in our study may not fully represent the changes that take place in vivo during JAK inhibitor therapy. We have previously shown that baricitinib and tofacitinib are rapidly taken up by RA neutrophils in cell culture where they not only prevent cytokine-induced phosphorylation of transcription factors, but also reverse cytokine-priming of ROS production in a little as 30 min [20]. Therefore their full effect on neutrophil metabolism should be evident after 2h incubation. Future studies should extend this work via a longitudinal study of people with RA pre- and post- oral administration of baricitinib and tofacitinib to confirm the in vivo effects on metabolism, ROS and NET production described in this work, and establish how this correlates to improvements in disease activity. Such clinical studies will also account for the effect of JAK inhibition on other elements of the immune system, which will likely impact neutrophil phenotype. The JAK-STAT pathway plays a role in development, proliferation, and function of T, B and NK cells. These cells produce and respond to cytokines including IL-2, IL-4, IL-7, IL-9, IL-15 and IL-21 which rely on JAK1-JAK3 activity. In the context of RA, the effects of JAK-STAT signalling include but are not limited to production of the chemokine IL-10 by in collagen-stimulated B cells which inhibits neutrophil recruitment [64] and IL-9 by Th-9 cells which prolong the survival of neutrophils in synovial fluid and increase MMP-9 production.”
Reviewer 2 Report
The Manuscript: „ Metabolic profiling of rheumatoid arthritis neutrophils reveals altered energy metabolism that is not affected by JAK inhibition’’ by Susama Chokesuwattanaskul is aimed towards determining the differences in the metabolome of neutrophils from healthy controls and individuals with RA as well as the effect of different JAK inhibitors on the metabolome of healthy and RA neutrophils. Works from the same group has previously also identified that migration of rheumatoid arthritis neutrophils towards interleukin-8 is prevented by JAK inhibitors and the priming of the respiratory burst is not prevented by JAK inhibitors. The present findings strengthen the previously established notions on repressing effects of JAK inhibitory drugs. The study is nicely conducted with elaborate description of methodology and documentation of subsequent result. After going through the manuscript, I have following comments to the author:
1. My biggest concern is the very small sample size of the study. I would suggest the authors to clarify whether the sample size of the study was large enough to interpret the statistical findings of the study.
2. Please provide the age and gender distribution of patients and controls for appropriate comparison of the outcomes.
3. Please provide a brief and to-the-point conclusion (something like take-home-message) of the study.
